# Investigation of the Role of *BMP2* and *-4* in ASD, VSD and Complex Congenital Heart Disease

**DOI:** 10.3390/diagnostics13162717

**Published:** 2023-08-21

**Authors:** Dimitrios Bobos, Giannoula Soufla, Dimitrios C. Angouras, Ioannis Lekakis, Sotirios Georgopoulos, Euthemia Melissari

**Affiliations:** 1Department of Pediatric Cardiothoracic Surgery, Onassis Cardiac Surgery Center, 17674 Athens, Greece; dimitriosbobos@gmail.com; 2Department of Hematology and Blood Transfusion, Onassis Cardiac Surgery Center, 17674 Athens, Greece; 3Department of Cardiac Surgery, Faculty of Medicine, Attikon University Hospital, National Kapodistrian University of Athens, 15771 Athens, Greece; 4Second Department of Cardiology, Attikon Hospital, Athens Medical School, National Kapodistrian University of Athens, 15771 Athens, Greece; 5First Department of Surgery, Laikon General Hospital, Medical School, National Kapodistrian University of Athens, 15771 Athens, Greece

**Keywords:** BMP2, BMP4, mRNA, protein, VSD, ASD, congenital heart disease

## Abstract

Congenital heart malformations (CHMs) make up between 2 and 3% of annual human births. Bone morphogenetic proteins (BMPs) signalling is required for chamber myocardium development. We examined for possible molecular defects in the bone morphogenetic protein 2 and 4 (*BMP2, -4)* genes by sequencing analysis of all coding exons, as well as possible transcription or protein expression deregulation by real-time PCR and ELISA, respectively, in 52 heart biopsies with congenital malformations (atrial septal defect (ASD), ventricular septal defect (VSD), tetralogy ofFallot (ToF) and complex cases) compared to 10 non-congenital heart disease (CHD) hearts. No loss of function mutations was found; only synonymous single nucleotide polymorphisms (SNPs) in the *BMP2* and *BMP4* genes were found. Deregulation of the mRNA expression and co-expression profile of the two genes (*BMP2/BMP4*) was observed in the affected compared to the normal hearts. *BMP2* and *-4* protein expression levels were similar in normal and affected hearts. This is the first study assessing the role of BMP-2 and 4 in congenital heart malformations. Our analysis did not reveal molecular defects in the *BMP2* and *-4* genes that could support a causal relationship with the congenital defects present in our patients. Importantly, sustained mRNA and protein expression of *BMP2* and *-4* in CHD cases compared to controls indicates possible temporal epigenetic, microRNA or post-transcriptional regulation mechanisms governing the initial stages of cardiac malformation.

## 1. Introduction

Congenital heart disease (CHD) appears in humans with high rates of morbidity and mortality (1). Congenital heart malformations (CHMs) are encountered in about 2–3% of annual human births and can be classified in simple or complex cases wherein more than one heart malformation is present. The incidence of moderate and severe CHD is estimated ~75/1000 live births worldwide [1,2,3,4]. Intervention is needed in 1% of patients, whereas adult patients frequently appear with a variety of cardiac complications, including coronary heart disease, arrhythmias and heart failure [5]. Although chromosomal aberrations and molecular defects have been found in sporadic cases, the genetic etiology of CHD remains to be fully elucidated [6,7].

The heart is the first formed organ during embryonic development. The areas designated to form the chambers of the heart are aligned with the atrioventricular canal through cardiac looping discriminating in this way the atrial and ventricular regions [8]. Cardiac remodelling comprises the next step of this process, leading to chamber septation and finally formation of the distinct left and right ventricles as well as the left and right atria. Primary heart tube separation into left and right components as well as septation of the atrial and ventricular chambers both comprise events taking place during the chamber formation. Fusion of the septums of the inter-atrial and inter-ventricular regions of myocardial origin with the atrioventricular septum of the endocardial cushion leads to ultimate chamber separation [8].

Overall heart formation is a procedure tightly controlled by gene regulation, transcription-factor-regulated signaling pathways, epigenetic regulation and microRNAs [9,10]. The role of transcription factors in heart development has been extensively studied in the past decade. Moreover, a variety of CHD-causing mutations has been identified to be associated with early heart development providing a number of possible causes of CHD [10,11]. However, the exact molecular mechanisms underlying the defective cushion and tissue type morphogenesis in the human heart leading to ventricular septal defect (VSD), atrial septal defect (ASD) or other complex congenital defects of the human heart remain unclear.

BMPs comprise growth factors with multiple functions that belong to the transforming growth factor-β (TGF-β) superfamily that has more than 30 ligands that have been recognized so far. Several subgroups are identified such as BMPs, growth and differentiation factors (GDFs), and activins/inhibins, TGFβs, etc. [12]. Tgf-β superfamily signalling takes place through a heterodimeric complex formed by type I and type II receptors. Initially, BMPs were found to induce bone and cartilage formation; further studies, however, have elucidated their significant role in embryogenesis and in a variety of functions, such as growth, differentiation and apoptosis of several types of cells, including chondroblasts, osteoblasts, neuronal cells, and epithelials cells [13,14]. The BMP signaling pathway involves BMP ligand binding to a type II receptor, for example BMPR2, leading to activation of a type I receptor, for example BMPR1A, resulting in the phosphorylation of Smads (R-Smad) signal transducers which take up the role of transcription inducers of downstream genes. Non-Smad signaling pathways (such as a MAPK-signaling pathway) also govern BMP signaling [15]. BMP signaling initially ensures endocardium activation by proper environment establishment, followed by EMT promotion and invasion of the mesenchyme into the cardiac cushions with the aid of Notch and TGFβ signalling.

*BMP-2* and *-4* have been established to possess multiple functions during the cardio genetic process, particularly at the initial stages of heart development. Recent research in animal models has revealed that functional disruption (knockout) of *BMP-2* and *-4* leads to nonviable mice, whereas animals with BMP deficiency exhibit major cardiac abnormalities [16,17]. Conditional deletion of the *BMP4* gene in cardiomyocyte in mice results in ventricular septal defects (VSD), atrioventricular canal defect (AVCD) and double outlet right ventricle (DORV) [18]. *BMP2* and *BMP4* compound heterozygous mice embryos also have VSD [19,20]. Analysis of BMP receptors further determined that BMP signaling is required for the development of a chamber myocardium. For example, conditional deletion of the *BMPR2* genein developing hearts in mice results in cardiac defects including DORV, VSD, and AV cushion defects [21]. Additional proof of the critical role of BMPs in the determination of atrial and ventricular fate has arisen from animal studies indicating that BMP activity is necessary for the correct differentiation of progenitor cardiac cells [15].

To provide some insight into the role of *BMP-2* and *-4* in ASD, VSD and other complex congenital defects, in the present study we examined 52 cases of congenital heart defects (ASD, VSD and complex defects) for possible molecular defects in the *BMP2* and *BMP4* genes. Therefore, we performed sequencing analysis of all coding exons of the *BMP2* and *BMP4* genesin all patients as well as in 100 non-CHD blood donors used as controls. Furthermore, we evaluated the mRNA expression and co-expression profile of *BMP2* and *BMP4* in both tissue biopsies from 52 defective hearts as well as 10 non-CHD hearts used as controls, in order to possibly identify transcriptional defects that may be associated with the congenital heart defects observed. Finally, *BMP2* and *-4* protein levels were assessed in both affected and normal tissue extracts with the purpose of revealing possible post-transcriptional regulation of the genes of interest in the setting of the congenital defects examined.

## 2. Materials and Methods

### 2.1. Patients and Controls

Tissue biopsies from 52 patients with congenital heart defects were collected during surgery at the Onassis Cardiac Surgery Center during a recruitment period of 2 years. Similarly, 10 tissue biopsies were collected from patients with non-CHD undergoing surgery for other heart conditions and were used as controls for the mRNA and protein expression analysis. The small number of non-CHD hearts is due to surgical reasons or the refusal of patients with other heart conditions to take part in this study. Samples were then immediately refrigerated at −80 °C before RNA and protein extraction.

Blood samples from all patients as well as 100 blood donors with no known CHD condition were collected and kept at 4 °C until DNA extraction.

All patients or relatives signed an informed consent form approved by the Onassis Cardiac Surgery Center Ethics Committee to participate in the study. The study design was in accordance with the Declaration of Helsinki Guidelines. Congenital defect cases included in the study are summarized in Table 1. Cardiac lesion cases that are studied herein other than VSD, ASD and Fallot, are categorized under complex/other defects and include cases such as atrioventricular septal defect (AVSD), coarctation of aorta and hypoplastic left heart syndrome.

Mean age of the patients is depicted in Table 2.

### 2.2. DNA Extraction and Sequencing

Whole genomic DNA was extracted from peripheral blood leukocytes using a QIAamp DNA blood mini kit (cat. no. 51104; Qiagen, Hilden, Germany). DNA star primer designing software was used to design primer pairs for PCR amplification of all exons of the *BMP2* and *BMP4* genes. The set of primers used and annealing temperatures are listed in Table 3. PCR reactions were performed with the use of GoTaq Flexi polymerase (Promega, Madison, WI, USA). PCR products were purified with a NucleoFast96 PCR Plate, 96-well ultrafiltration plate for PCR clean-up (Macherey Nagel REF 743100.10) and sequenced to determine possible molecular defects such as insertions/deletions or single nucleotide polymorphisms (SNPs). A Hardy–Weinberg equilibrium test was used for the CHD and control populations for each SNP identified.

### 2.3. Multiple Sequence Alignment

From the National Center for Biotechnology Information website (http://www.ncbi.nlm.nih.gov/) accessed on 16 September 2021, the *BMP-2* and *-4*protein sequences of various species were obtained. Using Vector NTI software (Suite 9; Thermo Fisher Scientific, Inc., Waltham, MA, USA), we performed multiple-sequence alignments of these proteins.

### 2.4. RNA Extraction, Reverse Transcription and Real-Time Quantitative PCR (RT-qPCR)

Total RNA was isolated from fresh tissue and homogenized with a power homogenizer using TRIzol reagent (Invitrogen, Carlsbad, CA, USA). The RNA concentration and purity were determined on a UV spectrophotometer (Biophotometer, Eppendorf) by absorbance measurements (260-nm absorbance and 260/280-nm absorbance ratio). The RNA integrity was examined by 1% agarose gel electrophoresis and ethidium bromide staining.

Reverse transcription reactions for the preparation of first-strand cDNA from 2 μg of total RNA were performed using the AffinityScript^TM^ Multi Temperature cDNA synthesis kit (Stratagene, La Jolla, CA, USA). Random hexamers were used as amplification primers. Real-time PCR reactions were performed using the Mx3000P real-time PCR system (Stratagene, La Jolla, CA, USA) with SYBR^®^ Green I Master Mix (Stratagene). Data were collected and analyzed using the Mx3000P real-time PCR software version 2.00, Build 215 Schema 60 (Stratagene). Glyceraldehyde-3-phosphate dehydrogenase (GAPDH), beta-actin and 18sRNA were used as internal controls to normalize the mRNA expression levels of the BMPs (*BMP-2* and *-4*) examined. The primer pair sequences used for quantitative real-time RT-qPCR are shown in Table 4. PCR products were analyzed by electrophoresis on 2% agarose gels, stained with ethidium bromide and photographed on a UV light transilluminator. The relative fold gene expression (*BMP2* and *BMP4* genes) of samples was calculated using the ΔΔCq method [22]. Procedures were repeated with a cDNA template and synthesized three times from the same RNA. The relative fold BMP-2 and -4 expression of each sample tested represents the mean value of data acquired from three independent RT-qPCR experiments. Reproducibility of the real-time PCR results for the same samples was 99%.

### 2.5. Protein Extraction, Quantification and Assessment

Protein was isolated from frozen tissue after homogenization with a power homogenizer using TRIzol reagent (Invitrogen, Carlsbad, CA, USA) according to the manufacturer’s instructions. Total protein quantification was performed using the Quick Start™ Bradford Protein Assay Kit (Biorad, Hercules, CA, USA). BMP2 and -4 protein levels in patients and control tissue extracts were determined with the Human *BMP-2* and *BMP-4* Immunoassay (Cat number: DBP200 and DBP400; R&D systems, Minneapolis, MN, USA).

### 2.6. Statistical Analysis

The one-sample Kolmogorov–Smirnov test was used to assess the normality of the distribution of relative fold *BMP-2* and *-4* expression values of the samples. Accordingly, relative fold gene expression was compared in the groups of patients and controls using non-parametric procedures. The Spearman rank correlation test (non-parametric) was employed to examine pair-wise correlations of relative fold *BMP-2* and *-4* expression in the defected and normal heart groups. Probability values or differences of less than 0.05 were considered to indicate statistically significant differences.

The Hardy–Weinberg equilibrium test was used for the CHD and control populations for each SNP identified. The statistical analyses were performed using χ^2^ tests (descriptive statistic crosstalk) to calculate the odds ratios and *p*-values, implemented using SPSS software (version 13.0; SPSS, Inc., Chicago, IL, USA). In addition, Online Encyclopedia for Genetic Epidemiology studies (OEGE; http://www.oege.org/software (accessed on 1 September 2022)) online software was used to perform the Hardy–Weinberg equilibrium test for the CHD and control populations.

Statistical calculations were performed using SPSS software, version 15.

## 3. Results

### 3.1. Sequencing

Sanger sequencing of all coding exons of *BMP2* and *BMP4* genes was performed on all 52 patients with a congenital heart disease, as well as on 100 blood donors with non-CHD hearts. No deletions/insertions, no loss-of-function mutations or other coding-sequence-affecting mutations were found. Only the following synonymous SNPs were observed: *BMP2* rs1049007(G > C) and rs235768(T > A), as well as *BMP4* rs17563(T > C), which were not bound to be significantly associated with the risk of congenital heart disease when compared to controls according to the previous study by Li et al. [23]. The results were in accordance with the Hardy–Weinberg equilibrium and although the study of Li et al. was performed in the Chinese population, it comprised at least four times more individuals and did not identify statistically significant associations between these genetic variations and the risk of CHD [23]. Similarly, our findings indicated the absence of the association of *BMP2* rs1049007 (G > C) and rs235768 (T > A), and *BMP4* rs17563(T > C) genotype variations with the risk of congenital heart disease (Pearson χ^2^ test, Asymptomatic *p* value (2-sided) = 0.624 rs1049007; *p* value = 0.742, rs235768; *p* value = 0.923, rs17563, degrees of Freedom (df) = 2 in all cases).

The *BMP-2* and *-4* protein sequences were found to be conserved during evolution because no differences in protein sequences were found when compared with the sequences of other species such as rodents, fish and birds. Comparison with protein sequences of other mammals such as Pan troglodytes, Macaca mulatta and Homo sapiens, provided similar results. Specifically, upon application of multiple-sequence alignment analysis high conservation of both the rs235768 and the rs1049007 variations was demonstrated resulting in no alteration the protein sequence. Furthermore, high conservation was found for the 190Ser and 152Val residues in *BMP2* and *BMP4* genes which are located in highly conserved regions of the proteins.

### 3.2. mRNA Expression Analysis Results

#### 3.2.1. Part A of mRNA Expression Analysis

In this part of the analysis, the normal sample with the highest Ct value, therefore the sample with the lowest gene expression, was selected as a calibrator/reference sample to calculate delta delta Ct.

##### Relative *BMP2* and *BMP4* Fold mRNA Expression Values in Normal and Defective Hearts

Similar relative fold *BMP2* and *BMP4* mRNA expression values were observed in the normal and CHD-affected hearts (*p* > 0.050 Mann–Whitney test) Table 5. After stratification of the CHD cases into subgroups of ASD, VSD, Fallot and other complex defects, suitable statistical tests were performed to correct for a small number of cases in each subgroup (Bonferroni). When the relative fold *BMP2* and *BMP4* expression levels were compared between the three subgroups (ASD, VSD, Fallot, complex/other), no statistically significant differences were observed (*p* > 0.050). Moreover, when each of the congenital heart defect subgroups (e.g., ASD) was compared to the normal hearts similar relative expression values were found.

The lack of statistically significant differences observed between the two groups of normal and CHD-affected hearts is likely to be expected since ventricular septal defect (VSD), atrial septal defect (ASD) or other complex congenital defects of the human heart comprise different entities with different morphological features and grouping them together may not be appropriate. However, stratification of the CHD cases into subgroups of ASD, VSD, Fallot and other complex defects failed to provide substantial differences between them or between them and the control group regarding the genes of interest. The small number of cases in each subgroup is to be taken into serious consideration when trying to assess our findings and maybe one cause of the absence of associations observed.

##### Pair-Wise Co-Expression Analysis of Relative Fold *BMP2* and *-4* mRNA Expression Values

Pair-wise co-expression analysis of relative fold *BMP2* and *-4* mRNA expression values was assessed separately in each group (CHD, Controls) as well as in each subgroup (ASD, VSD, Fallot, complex/other) if statistically allowed by the number of cases in each group/subgroup with the Spearman correlation test. The absence of statistically significant positive or negative correlations between relative fold *BMP2* and *-4* expression values was observed and this was determined by *p* > 0.050.

By this point, our findings seem to move away from the hypothesis that *BMP2* and *-4* mRNA may differ between normal and congenital defective hearts; however, there is a number of factors that could also affect our findings such as the type of analysis performed. Specifically, selection of the sample with the lowest gene expression as the calibrator/reference sample to calculate delta delta Ct may not be appropriate according to our belief. For this reason, we continued with another type of analysis below (part B of mRNA expression Analysis) in which we calculated the ”Average Ct” values of the control group in order to create a “Control average” which we used afterwards as a calibrator/reference sample to calculate delta delta Ct.

#### 3.2.2. Part B of mRNA Expression Analysis

In this analysis we calculated the ”Average Ct” values of the control group in order to create a “Control average” which we used afterwards as a calibrator/reference sample to calculate delta delta Ct. All results in this section are presented relative to the control average Ct values.

##### Relative *BMP2* and *BMP4* Fold mRNA Expression Values in Normal and Defected Hearts

Similar relative fold *BMP2* and *-4* mRNA expression values were observed in the normal and CHD-affected hearts (*p* = 0.19, *p* = 0.47, respectively, independent samples *t*-test). After stratification of the CHD cases into subgroups of ASD, VSD, Fallot and other complex defects, suitable statistical tests were performed to correct for a small number of cases in each subgroup (Bonferroni). When the relative fold *BMP2* and *-4* expression levels were compared between the subgroups (ASD, VSD), no statistically significant differences were observed (*p* = 0.50, *p* = 0.79, respectively, *t*-test). Moreover, when each of the congenital heart defect subgroups ASD and VSD was compared to the normal hearts, similar fold expression values were found regarding *BMP2* mRNA (*p* = 0.72, *p* = 0.33, respectively, *t*-test) and *BMP4* mRNA (*p* = 1.00, *p* = 0.99, respectively, *t*-test). The Fallot patient group was found to express significantly more *BMP2* mRNA than the control group (*p* < 0.001, *t*-test); however, similar *BMP4* mRNA expression levels were found between these two groups (*p* = 1.00, *t*-test). Upon comparison of the group of complex/other congenital abnormalities with the controls, a significant rise in the *BMP2 and BMP4* mRNA levels was observed (*p* < 0.001, in both cases, *t*-test); however, due to the heterogeneity of this group of congenital defects we cannot attribute this result to a specific congenital defect. We can only suggest that a large number of cases within this patient group may provide insight into the role of BMPs in this group.

##### Pair-Wise Co-Expression Analysis of Relative Fold *BMP2* and *-4* mRNA Expression Levels

Pair-wise co-expression analysis of relative fold *BMP2* and *-4* mRNA expression values was assessed separately in each group (CHD, Controls) as well as in each subgroup (ASD, VSD, Fallot, complex/other) if statistically allowed by the number of cases in each group/subgroup with the Spearman correlation test. In the control group, *BMP2* mRNA levels were found to be significantly correlated with the *BMP4* mRNA levels (*p*= 0.002, Spearman’s rho = 0.882). Interestingly, a statistically significant positive correlation (co-expression) was observed between relative fold *BMP2* and *BMP4* expression values in the group of VSD cases (*p* = 0.016, Spearman’s rho = 0.767). Absence of statistically significant positive or negative correlations between relative *BMP2* and *BMP4* expression values was observed in all other patient subgroups (*p* > 0.050, Spearman correlation). Of note, a similar co-expression profile of *BMP2* and *BMP4* was observed in the VSD cases as well as in the normal hearts, suggesting that perhaps this specific congenital abnormality may not be associated with transcriptional molecular defects of the genes studied. On the other hand, deregulation of the co-expression pattern of *BMP2* and *BMP4* was observed in all other patient subgroups (ASD, Fallot and complex/other cases), indicating that *BMP2* and *BMP4* may play a role in these settings.

### 3.3. BMP-2 and BMP-4 Protein Levels in Normal and Defective Hearts

*BMP-2* protein was occasionally expressed in the CHD cases, whereas it was not expressed at all in the normal heart biopsies. In detail, *BMP2* protein expression was observed in 6/12 VSD cases, in 2/7 ASD cases, in 2/10 Fallot and in 2/23 complex/other congenital defects. Overall, *BMP2* protein expression levels were not found to be significantly different in the group of CHD cases compared to controls (*p* = 0.16, *t*-test). Of note, a close look at the raw data indicates that *BMP2* protein is scarcely expressed in all cases with the exception of the VSD group that exhibited BMP2 protein expression in 50% of the cases.

*BMP4* protein levels did not differ significantly between the normal and CHD-affected hearts (*p* = 0.91, *t*-test). *BMP4* protein expression was observed in nearly all CHD cases as well as in controls. After stratification of the CHD cases into subgroups of ASD, VSD, Fallot and other complex defects, suitable statistical tests were performed to correct for the small number of cases in each subgroup (Bonferroni). Comparison of *BMP-2* and *-4* protein expression levels in the subgroups (ASD, VSD, Fallot, complex/other) failed to provide any substantial statistical differences (*p* > 0.050). Moreover, when each of the congenital heart defect subgroups (ASD, VSD, Fallot, Complex/other) was compared with the normal hearts, similar *BMP4* protein expression levels were found (*p* = 0.48, *p* = 0.31, *p* = 0.68, *p* = 0.64 respectively, *t*-test).

### 3.4. Pair-Wise Correlation of (a) BMP-2mRNA and BMP2 Protein and (b)BMP-4 2mRNA and BMP2 Protein Levels in Normal and Defective Hearts

In the CHD-affected hearts, *BMP2* mRNA and protein levels were not correlated (*p* = 0.11, Spearman rho = −0.202), whereas this statistical analysis could not be performed in the group of normal hearts because no *BMP2* protein expression was observed in that group. On the other hand, *BMP4* mRNA exhibited a significantly negative correlation with *BMP4* protein in the CHD group (*p* = 0.04, Spearman rho = −0.259), while no correlation was found between *BMP4* mRNA and protein levels in the normal hearts (*p* = 0.73, Spearman rho = 0.134). The latter means that in the CHD-affected hearts when *BMP4* mRNA is over-expressed, then *BMP4* protein is down-regulated and vice versa.

## 4. Discussion

*BMP-2* and *-4* are implicated in the formation of the heart from the overlying mesoderm [24,25]. BMP-2 can induce expression of early cardiac markers (NKX2-5 and GATA-4), within anterior mesodermal cells, in proximity to the heart-forming region [26,27]. Upon inhibition of BMP-2 and -4 expression, cardiac differentiation can be inhibited [26]. BMP-2 and -4 are therefore indispensable in cardiogenesis, and exert their function through their interaction with multiple extracellular signaling molecules [28]. Human embryonic stem cells (hESCs) require BMP signaling for the formation of different early cell types including the mesoderm [29], endoderm [30] and trophoblasts [31]. The role of *BMP2* and *BMP4* has also been demonstrated in the formation of the outflow tract [32].

BMP signaling requires BMP protein binding to heterotetrameric receptor complexes and receptor-mediated activation of Smad transcription factors [1]. BMPs are major players in valve development and the septation of cardiac chambers. *BMP2* derived from the myocardium produces the cushion extracellular matrix required for EMT by activating the expression of hyaluronic acid synthetase 2. BMP2 and BMP4 receptors hold various spatial roles. Interestingly, mice exhibiting interruption of the aortic arch, persistent truncus arteriosus, and missing semilunar valves have been found to have hypomorphic alleles of *BMPR2* [33].

The *BMP2* gene is located on chromosome 20p12.3. It consists of four exons and three introns, but the *BMP2* protein is encoded by exons 3 and 2. *BMP2* protein is indispensable for cardiac jelly formation [34] and absolutely *essential* for the initiation of the cardiac cushion formation [12,35]. Of note, upon BMP2 protein abolition in myocardium, the mature outflow tract (OFT) cardiac cushion development was not affected [33].

The *BMP4* gene is located on chromosome 14q22.2. It consists of four exons and three introns; however, the *BMP4* protein is encoded only by exons 3 and 4. *BMP4* protein expression has been demonstrated in the ventral splanchnic- and branchial-arch mesoderm and in the cardiac outflow tract myocardium [36]. Furthermore, BMP4 induces endocardial cushion formation by enhancing endocardial cells of the outflow tract to undergo epithelial–mesenchymal transition and invade the intervening space to form the cushions. Another important function of BMP4 is the cardiac neural crest induction towards invasion into the aortopulmonary septum and outflow tract cushions that are being formed [18]. Most importantly, it has been demonstrated in animal studies that upon *BMP4* inactivation within the myocardium the neonatal will die due to severe defects in septation and valve diseases such as ASD and VSD [37]. Recent studies have suggested that *BMP4* may exert its regulatory actions through a miRNA-mediated effector mechanism which in turn is responsible for the downregulation of cardiac progenitor genes leading to an increase in myocardial differentiation [38]. It is acknowledged that normal cardiac development suggests a complex procedure guided by high-precision signaling pathways and transcription factors with the goal of preserving the balance between proliferation, migration, and subsequent differentiation so as to avoid possibly fatal errors [39,40]. *BMP4* seems to be playing a crucial role in this process by regulating cardiac normal development through its interaction with multiple transcription factors.

Only a small number of studies has investigated CHD cases for possible genotype and allele distribution abnormalities of SNPs in the *BMP2* and *BMP4* genes, compared to normal hearts. Our study is the first according to our knowledge that examined all exons of the *BMP2* and *BMP4* genes to determine possible molecular defects such as insertions/deletions or single nucleotide polymorphisms (SNPs) that are possibly associated with the risk of congenital heart disease. Our findings, however, did not reveal any novel variations but confirmed in our cohort the presence of previously described SNPs *BMP2* rs1049007 (G > C) and rs235768 (T > A), and *BMP4* rs17563 (T > C) in the Chinese population. The 538T/C (rs17563) polymorphism of the *BMP4* gene has been found by Lin et al. to present significant differences in the genotype and allele distributions between patients with nonsyndromic cleft lip with or without a cleft palate (NSCLP) and control subjects in children of Chinese origin. Carriers of the 538 C allele were at significantly greater risk of NSCLP than the noncarriers [41]. Qian et al. suggested that the common *BMP4* intronic SNP rs762642this polymorphism may contribute to susceptibility to sporadic CHD in an additive model also in a Chinese population [42]. On the other hand, Li et al. provided evidence against statistically significant associations between these genetic variations and the risk of CHD [23]. Of note, all of the above research groups carried out their research in the Chinese population (Asian population) and their results cannot be extrapolated exactly to other ethnic groups such as the European population. Our findings, although extracted from a much smaller cohort of CHD patients, comprise the first evaluation of genotyping of all coding exons of *BMP2* and *BMP4* genes in CHD patients of the European population. Nevertheless, none of the SNPs found in our study displayed a significant association with CHD risk in accordance with the study of Li et al. We can only speculate that genetic variations in the *BMP2* and *BMP4* gene exons may not be associated with congenital heart defects. However, larger population-based, prospective studies are needed to clarify the impact of *BMP2* and *BMP4* polymorphisms on CHD susceptibility.

In addition, the present study comprises the first effort to provide new insights into the molecular basis of human CHD types (ASD, VSD, Fallot, complex/other) by investigating the *BMP2* and -4 mRNA and protein levels in a set of CHD cases. Our analysis revealed similar transcripts in CHD cases and control hearts with the exceptions of: (a) the group of Fallot cases that exhibited higher *BMP2* mRNA levels compared to controls and (b) the group of “complex/other” congenital defects that presented higher mRNA levels of both genes (*BMP2* and *BMP4*). Nevertheless, due to the heterogeneity of congenital defects in the group of “complex/other” cases, we cannot draw safe conclusions or even make speculations based on this finding. We can only comment that a larger set of biopsies that would adequately separate each congenital defect into distinct groups for statistical analysis may provide insight into the role of *BMP2* and *BMP4* mRNA in these settings.

*BMP2* protein was scarcely expressed in the CHD cases examined in our study, whereas it was totally absent in the control group of normal heart biopsies. We can only speculate that perhaps *BMP2* protein is essential in the initial steps of human heart development and that, after completion of septation and formation of the heart chambers, then *BMP2* protein does not have a reason for being expressed. *BMP4* protein expression, on the other hand, is demonstrated in both normal and CHD-affected hearts, indicating that possibly *BMP4* plays a constant and indispensable role in cardiac development as well as cardiac function. Taking into consideration the interaction of *BMP4* with multiple transcription factors in order to exert its functions, we can suggest that it is worth examining it further and in depth in order to elucidate its exact role in human heart development. The latter is reinforced by the deregulation of the co-expression profile of *BMP4* mRNA and protein that is observed in our study in CHD cases compared to controls.

Overall, taking into account the fact that both BMPs studied regulate cardiac normal development, which includes multiple signaling inputs and transcription factors, we can only suggest that perhaps temporal epigenetic, microRNA or post-transcriptional regulation mechanisms take place to orchestrate both the initial stages as well as the later stages of cardiac malformation in CHD in a time-dependent way. Further functional studies are needed to verify our hypothesis.

STUDY LIMITATIONS: The present study was conducted on a small number of human heart biopsies limiting the value of our findings. Both normal as well as VSD, ASD and Fallot cases were around 10 which can only give us a hint on whether it is worth undertaking a larger study to delineate the role of *BMP2* and *BMP4* in human heart development. Furthermore, cardiac lesion cases that are studied herein other than VSD, ASD and Fallot are categorized under complex/other defects and include cases such as atrioventricular septal defect (AVSD), coarctation of aorta and hypoplastic left heart syndrome. In this regard, our analysis of the group of complex/other defects is uninformative of the characteristics of each defect separately and we sought to identify a perhaps common feature of these defects in respect of BMP2 and BMP4 mRNA or protein expression.

## 5. Conclusions

In conclusion, the results of this study do not indicate any molecular defects in the *BMP2* and *-4* genes that could support a causal relationship with the congenital defects present in our patients. Importantly, sustained mRNA and protein expression of *BMP2* and *-4* in CHD cases compared to controls indicates possible temporal epigenetic, microRNA or post-transcriptional regulation mechanisms governing the initial stages of cardiac malformation.

## Figures and Tables

**Table 1 diagnostics-13-02717-t001:** Summary of congenital defects studied.

Congenital Defects	No of Cases
VSD	12
ASD	7
Fallot	10
Complex/other	23
Total No of cases	52

**Table 2 diagnostics-13-02717-t002:** Age of the patients.

Age	Years
Mean	5.7 years
Median	1.5 years
Range	1 day–44 years

**Table 3 diagnostics-13-02717-t003:** Primer sequences used for sequencing.

Growth Factor	Primer Pair Sequence (5′-3′)	Annealing (°C)
BMP2 exon2	AACGTTTGAGCTTCGGTCGGTCTT CTTTTGCCCTCATCTTCCCCATCA	60
BMP2 exon3A	GCTTCACCATTTCCCCCATTA CCCACCTGCTTGCATTCTGAT	61
BMP2 exon3B	TTCTTCCCTGTTTTTCTCTATCAA GACACCCACAACCCTCCAC	57
BMP4 exon1	GGGCGGCTGGAGGGGAGGATGT GCAACGCGTTCAGCCCAAGACC	66
BMP4 exon2A	GCCGGGTTCGAGCTGGGAGACG CAGGGGTGGTGAGGGCAGAGTGAA	66
BMP4 exon2B	GGCGGGCGCGGAGGTTGG CCGACGGAAGGGACAGC	64
BMP4 exon3	CTTTCCATCTTGCCCCTCCATTTC CACCTCCCCCTCTGTCTCCA	61
BMP4 exon4A	GCTTATTTTCCCCCAGTAGGTTTC GAGTGGCGCCGGGAGTTCTTATTC	61
BMP4 exon4B	CCTGGGCACCTCATCACACGACTA AGCGGCACCCACATCCCTCTACTA	66
BMP4 exon4C	GCGGGCCAGGAAGAAGAATAAGAA CAACAGGTGAGTGAAACAGAAGA	61

**Table 4 diagnostics-13-02717-t004:** Primer sequences used for Quantitative Real-time RT-qPCR.

Growth Factor	Primer Pair Sequence (5′-3′)	Annealing (°C)
BMP2 mRNA	CGTGTCCCCGCGTGCTTCTTAG CCTGCTGGGGGTGGGTCTCTGT	59
BMP4 mRNA	CCGCCCGCAGCCTAGCAAGAGT TAAAGAGGAAACGAAAAGCAGAGT	137
GAPDH	GGAAGGTGAAGGTCGGAGTCA GTCATTGATGGCAACAATATCCACT	60
b-actin	CGGCATCGTCACCAACTG GGCACACGCAGCTCATTG	60
18S RNA	AGTCCCTGCCCTTTGTACACA GATCCGAGGGCCTCACTAAAC	60

**Table 5 diagnostics-13-02717-t005:** *BMP2* and *BMP4* mRNA expression in CHD cases and controls.

	Control *	All CHD *	VSD	ASD	Fallot	Complex/ Other	*p* *
BMP2	1.78 ± 0.50	3.22 ± 0.37	1.66 ± 0.62	4.36 ± 0.76 **	2.77 ± 0.80	2.86 ± 0.65	0.19
BMP4	1.40 ± 1.31	1.84 ± 0.26	0.90 ± 0.22	0.87 ± 0.25	1.40 ± 0.53	1.99 ± 0.52	0.47

Results are expressed as mean ± SEM (standard error of the mean). * *p* > 0.050, Nonsignificant statistical differences were observed between CHD cases and controls. ** In Table 5, it seems that the mean *BMP2* mRNA expression in ASD cases (4.36 ± 0.76) is higher than the controls (1.78 ± 0.50). However, a close look at the raw data revealed that this is due to a single case exhibiting an extremely high value of *BMP2* mRNA expression that we considered being likely an artifact. When this extreme value was excluded, then the mean expression values ± SEM (standard error of the mean) were in fact similar to the controls (1.87 ± 0.67).

## Data Availability

Not applicable.

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
