# Peer review of "Investigation of the Role of BMP2 and -4 in ASD, VSD and Complex Congenital Heart Disease"

_diagnostics, 2023, doi:10.3390/diagnostics13162717_

Round 1

Reviewer 1 Report

Major comments:

1. In multiple sub-sections in the results the authors report findings from their experimental analyses. However the original data supporting these claims are not shown. For example, in section 3.2.1.2 there was a pair-wise co-expression comparison performed and the authors conclude that there was no association. These results are not shown in the manuscript, which makes it impossible to assess the impact of this finding.

2. Overall, the results section would benefit from being re-framed to include the author interpretation for each sub-section. The absence of interpretation of the findings takes away from the potential impact this work has in the field of CHD.

Minor comments:

1. Multiple abbreviations are not described the first time they appear in the manuscript (VDS, ASD, BMP to name a few).

2. In Methods 2.1, the authors describe that following tissue collection the samples were stored in -80oC. In 2.5, the protein extraction protocol states the use of fresh tissue. This is confusing and should be clarified.

3. It is unclear whether lines 183-185 should be included in the manuscript as it reads as it is meant to be used as instructions to authors.

4. In section 3.1 (lines 190-193), the authors mention two identified SNPs that "were not bound to be associated with risk for congenital heart failure...". Is this a result of this study? If so, the statistical analysis supporting this lack of association is missing. If not, the appropriate reference should be used. Please clarify.

5. Line 194-196: Please consider editing this paragraph to provide more rationale as to why this was performed, and how these results were obtained.

6. Some syntax (commas, etc) and grammar issues were present. Please consider proof-reading the manuscript to include commas, where appropriate.

The manuscript is well written and can be understood by native and non-native English speakers. Minor grammatical and syntax errors are observed that can be easily addressed via proofreading.

Reviewer 2 Report

I reviewed the manuscript on “Investigation of the Role of BMP2 and 4 in ASD, VSD and Complex Congenital Heart Disease”. The study did not find molecular defects in the BMP2 and-4 genes that could support a causal relationship with the congenital defects present but possibly temporal epigenetic, microRNA or post-transcriptional regulation mechanisms governing the initial stages of cardiac malformation as sustained mRNA and protein expression of BMP2 and -4 in CHD cases compared to controls. This manuscript needs to be improved, especially in the methodology and result presentations.

Introduction: suggest correcting the use of Bmp2 and Bmp4 in the certain paragraph

Methodology: Please state the sample size calculation and why reason only ten non- CHD heart as a control. It is good to give details about the underlying cardiac lesions categorised under complex defects Statistical analysis(lines 129-134) under DNA extraction and sequencing should be written under the Statistical analysis heading. 

Results: Should focus on the results rather than incorporating the statistical analysis in the results. Table 5 can be improved by putting the control and type of CHD in the column and BMP2 and BP4 in the row. Suggest having 2 decimal points for the p-value.

Discussion:  should discuss the findings of this study rather than focusing on literature review on BMP2 and -4. Please include the study limitation

Funding: Please remove the repeated words

Round 2

Reviewer 1 Report

The authors have sufficiently address my concerns.